# Thermodynamic Theory of Macrosystems: Entropy Production as a Metric

**DOI:** 10.3390/e27111136

**Published:** 2025-11-05

**Authors:** Sergey Amelkin

**Affiliations:** Institute of System and Software Engineering and Information Technology, National Research University of Electronic Technology, 124498 Moscow, Russia; amelkin@ist.education

**Keywords:** thermodynamic processes, macrosystem, entropy production, extensive variables, intensive variables, measure space

## Abstract

The article considers the description of a macrosystem in terms that do not depend on the nature of the macrosystem. The results obtained can be used to describe macrosystem models of thermodynamic processes, and to create interdisciplinary models that take into account interactions of various natures. The macrosystem model is based on its representation in the form of a self-similar oriented weighted graph where the equation of state is fulfilled for each node, which connects extensive variables. One of the extensive variables is entropy, the maximum of which corresponds to the state of equilibrium. For processes in which fluxes are linearly dependent on driving forces, Onsager’s relations are shown to be true, which makes it possible to prove that in the space of stationary processes, entropy production in a closed macrosystem is a metric similar to the Mahalanobis metric, which determines the distance between processes. Zero in such a space indicates reversible processes, and thus the production of entropy shows the degree of irreversibility as the distance from a researched process to a reversible one.

## 1. Introduction

Thermodynamics has historically been the first and main research focus of complex systems [1]. It is thermodynamic analogies that are used in the description of other systems such as economic, social, informational, and algorithmic systems when control is only possible in an averaged sense. We mean *averaged* because it is impossible to observe or control the state and behaviour of individual elementary particles, whose large aggregate forms a complex system.

In a thermodynamic system, the elementary particles are molecules, even though only statistical mechanics need discrete particles as a concept. The reason is that phenomenological (i.e., experimentally observable) patterns related to the state of a thermodynamic system are carried out through dynamic or statistical averaging of the interactions between individual molecules. However, such averaging is only possible under a priori assumptions about the nature of such interactions as they cannot be verified due to the incredibly large number of molecules in the system and their extremely small size, which makes them neither observable nor controllable [2].

The validity of such a framework is supported by the existence of phenomenological laws. However, in this case, the assumptions about the properties of individual molecular interactions must be derived from the observations of thermodynamic systems as they cannot serve as proof of the truth of the phenomenological laws per se.

The approach suggested in this work is the reverse one, which is based on a structural induction where the basic point is any scale level at which the properties of the system can be experimentally verified. This presupposes the fulfilment of the self-similarity condition, which is generally defined as follows: a set X is self-similar if there exists a finite set *K* indexing a collection of injective and non-surjective mappings ${\left\{\phi_{\kappa }\right\}}_{\kappa \in K},$ such that $X={⋃}_{\kappa \in K}\phi_{\kappa }(X)$ [3]. The self-similarity condition is actively used in thermodynamics [4], even though it is strictly fulfilled only when the set of subsystems whose union forms the given thermodynamic system is a continuum. This assumption is too strong for real thermodynamic systems but can be considered a good approximation.

The approach considered below is not limited to describing thermodynamic systems alone, but is intentionally extended to a more general representation. This allows for the consideration of a class of macrosystem models that includes thermodynamic systems.

## 2. Definition of Macrosystems

When analyzing patterns that arise in the exchange processes in systems of various natures (e.g., physical, chemical, economic, informational, and social), it is often advisable to use a macrosystem approach.

Macrosystems are systems in which the following conditions are satisfied:
Macrosystems consist of a large number of elementary objects, and this number is so large that any macrosystem can be considered as a continuum and can be divided into any finite number of subsystems, including those sufficient to determine the statistical characteristics of any given accuracy; each of the subsystems can be considered as a macrosystem. Any macrosystem Y is a nonempty collection S of pairwise disjoint subsets of Y closed under complement, countable unions, and countable intersections. S is a σ-algebra and ordered pair (Y, S) is a measurable space. The macrosystem Y can be considered as a union of a finite number of a lower level macrosystems $Y_{i}$: $Y={⋃}_{i\in Y}Y_{i}$. This condition is called the self-similarity condition [5].The macrosystem state is determined by the vector Q of state variables. It is assumed that the state variables satisfy the conservation law; therefore, we can consider vector Q as a vector of extensive variables. Because of non-negativity and countable additivity, vector Q is a vector measure and (Y, S,Q) is a measure space. In thermodynamic macrosystems, the extensive variables are internal energy, mol number, and volume [6], which the macrosystem can exchange with its external environment [7]; we will also consider the external environment as a set of macrosystems (and a higher-level macrosystem is the union of the macrosystem and its external environment). In the process of interaction between the subsystems *X* and *Y*, the values of the vectors $Q_{X}$ and $Q_{Y}$ change over time. In this way, exchange processes and their corresponding fluxes are formed, which are understood as the rates of change of the extensive quantities. We will denote fluxes between subsystems X and Y as $q_{XY}$.It is impossible to control each elementary object due to their extremely large number [7]. The macrosystem control can be organized only by impact on the parameters averaged over a set of elementary objects, namely–Changes in the parameters of the macrosystem’s external environment, the interaction with which determines the change in the values of extensive variables in the macrosystem;–Change in the values of extensive variables (for example, their extraction) in the macrosystem due to external interventions;–Changes in the characteristics of the exchange infrastructure to accelerate or, conversely, slow down the exchange processes.

## 3. Representation of the Macrosystem as a Graph

The macrosystem can be represented as a self-similar oriented weighted graph, the nodes of which correspond to the subsystems and the macrosystem’s external environment, and the edges correspond to the fluxes between the subsystems and between the subsystems and the macrosystem’s external environment. Each graph node (each subsystem) is characterized by a vector $Q=\left(Q_{0},\ldots ,Q_{N}\right)$, and fluxes between subsystems can be functionally unrelated to each other.

*Self-similar*—each graph node can be represented as a graph that describes the interaction of subsystems that form this node with each other and with their environment. Let us consider two interacting (the vector of the fluxes of extensive variables is $q_{XY}$) disjoint macrosystems X and Y. Due to the self-similarity condition, we can introduce sets $X={⋃}_{i}X_{i},Y={⋃}_{j}Y_{j}$ of pairwise disjoint subsystems corresponding to these macrosystems. Each subsystem is characterized by its own values of extensive variables: $Q_{i}$ for $X_{i}$ and $Q_{j}$ for $Y_{j}$. Flux $q_{ij}$ is formed between subsystems from different sets X and Y. In this notation, the following equations are valid:(1)$$\left.\begin{array}{c} \sum_{i\in X}Q_{i}=Q_{X};\sum_{j\in Y}Q_{j}=Q_{Y};\\ \sum_{\begin{array}{c} i\in X\\ j\in Y \end{array}}q_{ij}=q_{XY}. \end{array}\right\}$$

*Oriented*—the direction of each edge determines the positive sign of each flux; thus, if the edges are directed from node X to node Y, then(2)$$q_{XY}=-\nabla Q_{X}=\nabla Q_{Y}.$$

*Weighted*—the weight of the edge shows the flux intensity: the positive value of the ν-th flux $q_{XY,\nu }>0$ if the real flux of the extensive variable is directed towards the edge, and $q_{XY,\nu }<0$ if it is opposite to the edge’s direction. In addition, a matrix A of infrastructure coefficients is determined for all fluxes between nodes X and Y (that is, for all fluxes $q_{XY}$). This matrix is the metadata for the edges connecting nodes X and Y, and determines both the facility of exchange process and complementary or substitutionary features of the extensive variables [8].

Figure 1 shows an example of the self-similar oriented weighted graph of a part of a macrosystem.

## 4. Equilibrium State of the Macrosystem

Let us assume that two macrosystems X and Y exchange the vector of the extensive variables Q. At each moment of time t, the reserves of the extensive variables describing the state of macrosystems are equal to $Q_{X}(t),\;Q_{Y}(t)$. We represent macrosystems as a union of a finite set of subsystems: X and Y, respectively. An equilibrium state shall be a state $\left(Q_{X},\;Q_{Y}\right)$ in which the sum of fluxes at each moment of time t for each ν=0,…,N(3)$$q_{XY,\nu }(t)=\sum_{\begin{array}{c} i\in X\\ j\in Y \end{array}}q_{ij,\nu }(t)=0.$$
Thus, the equilibrium in the macrosystem is considered dynamic. The equilibrium condition can be defined as follows: at the any level of sets of subsystems $X,Y:X={⋃}_{i\in X}X_{i};Y={⋃}_{j\in Y}Y_{j}$, a vector of fluxes $q_{ij}\left(i\in X,j\in Y\right)$ is described by a time-independent distribution $f(\tilde{q})$ with expectation $E\left[\tilde{q}\right]=0$.

Note that the fluxes $q_{ij}(t)$, $\left(i\in X,j\in Y\right)$, and the random variable $\tilde{q}$, which describes the subsystem fluxes distribution, are vectors. At this level of subsystems, we can assume that a large number of factors affects the distribution of fluxes, which means that the distribution $f_{q}\left(\tilde{q}\right)$ can be described by a multivariate normal distribution. The parameters of this distribution are the expectation $E\left[\tilde{q}\right]$, which determines fluxes at the level of subsystems X and Y in accordance with (3), and the covariance matrix $Cov\left[\tilde{q}\right]$.

Assume that fluxes $q_{XY}$ arise due to the activity of some driving forces, which we can also consider at the subsystem level as a random vector, such that

–Fluxes $q_{XY}$ are linearly dependent on the exchange driving forces $\varphi_{XY}$ [6]: $q_{XY}=A\varphi_{XY}$—under this assumption and suggesting that the matrix of infrastructure coefficients A is constant, the driving forces distribution is also normal;–The covariance matrix $Cov\left[\tilde{q}\right]$ of subsystems of the X∪Y macrosystem depends on the driving forces intensity $\varphi_{XY}=E[\tilde{\varphi }]$ causing these fluxes, so that the matrices $Cov\left[\tilde{q}\right]$ and $Cov[\tilde{\varphi }]$ are jointly normalizable (their eigenvectors coincide)—due to the linear relationship between fluxes and driving forces;–Limits of correlation coefficients corresponding to the covariance matrix $Cov\left[\tilde{q}\right]$ for any ν,κ=0,…,N: $\underset{\varphi_{\mathrm{XY}}\to 0}{\lim}\rho_{\nu \kappa }=0,\underset{\varphi_{\mathrm{XY}}\to \infty }{\lim}\rho_{\nu \kappa }=1$—due to the redistribution of the extensive variables over a variety of subsystems, depending on the number of intermediate nodes in the graph chain to the contact point.

If the equilibrium condition is satisfied for a macrosystem X when interacting with each macrosystem from its environment, then such a macrosystem is called closed. If, for a macrosystem X, all its subsystems are in equilibrium when interacting with each other (but not necessarily with the environment of the macrosystem X)(4)$$\forall t,\forall i\in X:\sum_{j\in X}q_{ij}(t)=0,$$
then we can say that this macrosystem is in a state of internal equilibrium. For a macrosystem in internally equilibrium, all fluxes can be observed only at the boundaries of the macrosystem and its environment.

## 5. Extensive and Intensive Variables

Let us assume that two macrosystems X and Y exchange the vector of the extensive variables Q. At each moment of time t, the values of the extensive variables describe the state of macrosystems.

The macrosystem shall be described by a set of extensive and intensive variables:

–Extensive variables are such that for any two disjoint macrosystems X and Y (not necessarily in equilibrium):
(5)$$Q_{X\cup Y}=Q_{X}+Q_{Y};$$
Due to conservation law, all the values of the extensive variables are extensive variables;–Intensive variables v are such that for any two systems X and Y that are in equilibrium:
(6)$$v_{X\cup Y}=v_{X}=v_{Y}.$$


Extensive variables satisfy the neutral scale effect condition: if all extensive variables in all subsystems of the macrosystem are increased by n times, then the flux intensities in the macrosystem will not change. In particular, such a proportional increase in the extensive variables will not bring the macrosystem out of the internal equilibrium state if before that, the system was in internal equilibrium. The neutral scale effect condition provides the self-similarity of the macrosystem.

As a consequence of (6), intensive variable dependencies on the extensive variable values that determine the macrosystem state should be homogeneous functions of the zero-order.

## 6. Entropy of the Macrosystem

The set of extensive variables describes the macrosystem state. Among the extensive variables, we single out the value $S=Q_{0}$, which characterizes the objective function of the system. S and other extensive variables are functionally related: S=S(Q), $Q=\left(Q_{1},\ldots ,Q_{N}\right)$. We call this equation the system state equation.

The choice of S as the objective function is determined by the Levitin–Popkov Axioms [9], which impose the following conditions on this variable:For a controlled system, S = S(Q) given a fixed deterministic control v. Its stochastic state, which is characterized by a vector flux, is transformed into a deterministic vector Q(v), called the steady or stationary state, which belongs to a permissible set D(v).For any fixed vector $v\in D\left(v\right)$, there exists a vector $p\left(v\right)$ of a priori probabilities for the distribution of fluxes in the system $S\left(Q\right)$, such that the stationary state $Q\left(v\right)$ of the macrosystem under that given fixed control v is the optimal solution to the entropy-based optimization problem: $Q\left(v\right)=\;z\left[v\right]\left(p\left(v\right)\right)$, where $z\left[v\right]\left(p\left(v\right)\right)$ is the entropy operator defined as

(7)$$z\left[u\right]\left(p\right)=argmax\{S\left(p,Q\right):v\in D(v)\}.$$
Thus, the pair (p(v), Q(v)) simultaneously provides both the required vector of prior probabilities and the corresponding stationary state vector.

3.There exists an inverse mapping p = ζ[v](Q) such that the desired pair (p(v), Q(v)) is the unique solution to the system of the relations:
(8)$$\left\{\begin{array}{c} Q\left(v\right)=z\left[v\right]\left(p\left(v\right)\right),\\ p=\zeta \left[v\right]\left(Q\right). \end{array}\right.$$


Within the framework of the considered model, these statements can be interpreted as follows:
For any fixed deterministic control v=const, there exists a stable steady state $Q^{*}(v)$.The stable steady state $Q^{*}(v)$ is a state of an internal equilibrium corresponding to the maximum of $S\left(Q\right)$.The stable steady state $Q^{*}(v)$ is unique.

For a closed system, the condition S(Q)→max determines the spontaneous direction of exchange of extensive variables. However, a question arises about the homogeneity of the function $S\left(Q\right)$: entropy is a homogeneous function of degree one only in the systems that are in a state of internal equilibrium. In cases where the distribution p(v) is scale-invariant but does not correspond to internal equilibrium, a fractal structure of the macrosystem is observed, in which $S\left(Q\right)$ is a homogeneous function with a degree of homogeneity less than one. Taking this remark into account, entropy can still be considered an extensive quantity. The question to use the degree of homogeneity $S\left(Q\right)$ as a measure of equilibrium of the macrosystem should be given further consideration.

Since S is an extensive variable in the condition of internal equilibrium, then $S\left(Q\right)$ is a homogeneous function of the first-order: when scaling the system by n times or combining n identical macrosystems in the equilibrium state:(9)nS=S(nQ).
In accordance with the Euler relations for homogeneous functions(10)$$S\left(Q\right)=Q\nabla S=\sum_{\nu =1}^{N}Q_{\nu }\frac{\partial S}{\partial Q_{\nu }}.$$

Let us denote $v_{\nu }=\partial S/\partial Q_{\nu }$. Since $S\left(Q\right)$ is a homogeneous function of the first-order, then $v_{\nu }(Q)$ $\left(\nu =1,\ldots ,N\right)$ are homogeneous of the zero-order, i.e., they are intensive variables: for any n value $v_{\nu }\left(nQ\right)=v_{\nu }(Q)$. This means that a proportional increase in the extensive variables will not bring the macrosystem out of the internal equilibrium state if before that, the system was in internal equilibrium.

Under the assumption that the function $S\left(Q\right)$ is differentiable and its partial derivatives are continuous, in accordance with the necessary condition for the function to be differentiable, there exists a total differential(11)$$dS=\sum_{\nu =1}^{N}\frac{\partial S}{\partial Q_{\nu }}{dQ}_{\nu }=\sum_{\nu =1}^{N}v_{\nu }{dQ}_{\nu }.$$

Two consequences of Equation (11) can be formulated.

**Consequence 1.** By differentiating the Euler relation (10), we obtain(12)$$dS=\sum_{\nu =1}^{N}\left(v_{\nu }{dQ}_{\nu }+Q_{\nu }{dv}_{\nu }\right).$$
By comparing (11) and (12), we see that the second term in (12) should be equal to zero:(13)$$\sum_{\nu =1}^{N}Q_{\nu }{dv}_{\nu }=0.$$

**Consequence 2.** The exchange process between subsystems X and Y (with a positive direction of fluxes from X to Y) can be described using Equations (2) and (11) as follows:(14)$$\frac{dS}{dt}=\frac{dS_{Y}}{dt}+\frac{dS_{X}}{dt}=\sum_{\nu =1}^{N}\left(v_{Y\nu }-v_{X\nu }\right)q_{XY,\nu }.$$

The parameter S is an extensive variable and its dependence on other state variables S(Q) is an objective function for spontaneous processes in the macrosystem.

From the established relations, the following conclusion can be drawn:

Let Q be a vector measure on a measurable space (Y, S,Q). Then, the entropy of a subsystem X∈Y is defined as a function $S_{X}=S(Q_{X})$, such that

–$S_{X}\leq S(Q^{*})$, where $S(Q^{*})$ is the entropy of *X* under internal equilibrium of all subsystems *X*;

–$S\left(Q_{X}\right)+S\left(0\right)=S(Q_{X})$, where $S\left(0\right)$ is the entropy of a subsystem with a zero-vector measure;

–For any two subsystems $X_{1},X_{2}$ such that $Y=X_{1}\cup X_{2}$, $Q_{X_{1}\cap X_{2}}=0$, it holds that $S_{Y}=S_{X_{1}}+S_{X_{2}}$.

These properties correspond to Shannon–Khinchin Axioms for entropy [10].

The entropy function meets the conditions of Shannon entropy if the number of subsystems is large enough. S statistically corresponds to the entropy of the distribution p(v(Q)) of state parameters of subsystems of the macrosystem. Let us explain this statement. Since there are no restrictions on the possible random vector values with a finite variance, the normal distribution corresponds to the maximum entropy value (this can be considered as a justification for the distribution law of subsystems parameters that form a macrosystem).

The entropy of the normal distribution consists of two terms: $S=S_{0}+0.5\ln\det R$, where $R=\left(\rho_{\nu \kappa },\;\nu ,\kappa =1,\ldots ,N\right)$ is the correlation matrix. The first addend corresponds to the complete independence of the system elements and characterizes the structure of the macrosystem, and the second describes the relationships in the macrosystem. The correlation $\rho_{\nu \kappa }$ (ν,κ=1,…,N) between fluxes increases, which is observed with the growing magnitude of the exchange driving forces $\varphi_{XY}$. Figure 2 illustrates the kind of dependency of $\rho_{\nu \kappa }\left(\varphi_{XY}\right)$ when components $\varphi_{\nu }$ and $\varphi_{\kappa }$ of the vector $\varphi_{XY}$ are changed. The second term of entropy expression, which is always negative at R≠E, (E is the identity matrix), decreases. This explains the statement about S as an objective function that reaches its maximum when the macrosystem reaches an equilibrium state, and a state function that sets the direction of processes in a closed system, increasing the dispersion of low-level subsystem parameters while simplifying the macrosystem structure at a high level.

In the process of spontaneous, not forced exchange, the S value of a closed macrosystem cannot decrease since $S\left(Q\right)$ is the objective function for the system behaviour. In order for the value σ=dS/dt, called the entropy production, to be positive during the any process of given non-zero intensity, it is sufficient to fulfil the condition(15)$$sign\left(v_{Y\nu }-v_{X\nu }\right)=sign\left(q_{XY,\nu }\right)\forall \nu =1,\ldots ,N.$$

If the process duration is infinitely long, then the macrosystem’s natural evolution leads it to a state of internal equilibrium, which corresponds to the achievement of the maximum S(Q) value. This is equivalent to the statement that in the internal equilibrium state, S(Q) is maximum.

Intensive variables v=∇S can be considered as specific potentials. Their difference $\varphi_{XY}=v_{Y}-v_{X}$ according to (15) is the driving force of the exchange process. The fluxes are directed from subsystems with lower values of intensive parameters to subsystems with higher values of intensive parameters. Thus, Equation (14) can be rewritten as follows:(16)$$\sigma (v_{X},v_{Y})=\sum_{\nu =1}^{N}\varphi_{\nu }\left(v_{X},v_{Y}\right)q_{XY,\nu }\left(A,\varphi \left(v_{X},v_{Y}\right)\right),$$
where A is the matrix of infrastructure coefficients. Note that the infrastructure of the exchange process involves a wide variety of features of the subsystems’ boundaries medium. For example, in thermodynamic macrosystems these features are surface area, roughness, etc., all these parameters can be controls in the corresponding optimization problems, but here, A is assumed to be constant. If the driving force during the process is constant, then such a process is called stationary. Since $\varphi \left(v_{X},v_{Y}\right)$ is a linear relation and therefore the superposition principle is being applied, we single out the classes of reversible ($\varphi \left(v_{X},v_{Y}\right)\to 0$) processes and processes of minimal dissipation $\sigma \left(\varphi \right)\to \underset{\varphi }{\min}\;\left|\;q_{XY}\left(\varphi \right)=fix\right.$.

## 7. Differential Form of the State Equation

The state function can be given in differential form, as is typical for thermodynamic systems:(17)$$\delta \Phi =\sum_{\nu =1}^{N}F_{\nu }\left(Q\right)dQ_{\nu }.$$

If we cannot write a state function explicitly, then the problem of integrability of Φ(Q) should be solved. Equation (17) is the Pfaffian form. A Pfaffian form is said to be holonomic if there exists an integrating multiplier w(Q) such that(18)$$w\left(Q\right)\delta \Phi =\sum_{\nu =1}^{N}\frac{\partial S}{\partial Q_{\nu }}dQ_{\nu }=dS,\;where\;\frac{\partial S}{\partial Q_{\nu }}=w\left(Q\right)F_{\nu }\left(Q\right),\nu =1,\ldots ,N.$$

The Pfaffian form of two independent variables is always holonomic, that is, there is always an integrating multiplier w(Q). However, for N>2, the integrating multiplier exists if the holonomy conditions are satisfied: for any three different κ,μ,ν,(19)$$F_{\kappa }\left(Q\right)\left(\frac{\partial F_{\mu }}{\partial Q_{\nu }}-\frac{\partial F_{\nu }}{\partial Q_{\mu }}\right)+F_{\mu }\left(Q\right)\left(\frac{\partial F_{\nu }}{\partial Q_{\kappa }}-\frac{\partial F_{\kappa }}{\partial Q_{\nu }}\right)+F_{\nu }\left(Q\right)\left(\frac{\partial F_{\kappa }}{\partial Q_{\mu }}-\frac{\partial F_{\mu }}{\partial Q_{\kappa }}\right)=0.$$

These conditions are obtained from the equality of the second mixed derivatives with respect to any variable pairs (Maxwell’s relations)(20)$$\frac{\partial^{2}S}{\partial Q_{\nu }\partial Q_{\kappa }}=\frac{\partial \left(w(Q)F_{\nu }(Q)\right)}{\partial Q_{\kappa }}$$
and by exclusion of the integrating factor w(Q) from these equalities. In addition to conditions (19), it is necessary that all products $w(Q)F_{\nu }(Q)$ be homogeneous of zero-order homogeneity.

## 8. Concavity of Entropy Function

The $S\left(Q\right)$ function gradient determines the intensive variable vector of the macrosystem. As the extensive variables increase, the S value also increases, but at a slower rate (the diminishing returns law), so that for all =1,…,N ${\partial S/\partial Q_{\nu }=v}_{\nu }>0,\partial^{2}S/\partial Q_{\nu }^{2}=\partial v_{\nu }/\partial Q_{\nu }<0$. Moreover, the Hessian matrix $H_{S}=\left(\partial^{2}S/\left(\partial Q_{\nu }\partial Q_{\kappa }\right)\right)$ for homogeneous functions of the first-order homogeneity is negative semi-definite. Indeed, it follows from the Euler relations (10) that differentiating both parts of this equation per $Q_{\kappa }$, (21)$$\forall \kappa =1,\ldots ,N:\sum_{\nu =1}^{N}Q_{\nu }\frac{\partial^{2}S}{\partial Q_{\nu }\partial Q_{\kappa }}=0.$$

For an arbitrary vector x, the necessary conditions for the extremum of a quadratic form in x are(22)$$x^{T}H_{S}x=\sum_{\nu =1}^{N}x_{\nu }^{2}\frac{\partial^{2}S}{\partial Q_{\nu }^{2}}+\sum_{\nu =1}^{N-1}\sum_{\kappa =\nu +1}^{N}{2x}_{\nu }x_{\kappa }\frac{\partial^{2}S}{\partial Q_{\nu }\partial Q_{\kappa }}\to \underset{x}{\max}\;.$$(23)$$\frac{\partial {(x}^{T}H_{S}x)}{\partial x_{\nu }}=\sum_{\kappa =1}^{N}x_{\kappa }\frac{\partial^{2}S}{\partial Q_{\nu }\partial Q_{\kappa }}=0,$$
which corresponds to the maximum point (according to the diminishing returns law) of the quadratic form $x_{\nu }=Q_{\nu },\nu =1,\ldots ,N$. Since the product $Q^{T}H_{S}$ (21) is equal to zero, then $x^{T}H_{S}x$ at the maximum point is also equal to zero. Thus, for any value of $x:x^{T}H_{S}x\leq 0$, which has to be proved. Note that since the Hesse matrix is symmetric, then all its eigenvalues are real numbers.

The negative semi-definiteness of the Hessian matrix $H_{S}$ corresponds to the upward convexity (concavity) of the S(Q) function and S unimodality as the objective parameter. As the reserves of the extensive variables in the macrosystem increase, S increases, and the intensive parameters decrease, reducing the magnitude of the resource exchange driving force. In accordance with (14), this behaviour of intensive parameters leads to the fact that when the macrosystem is affected, changing its internal equilibrium conditions; the resource exchange processes are directed towards counteracting changes, and thus the Le Chatelier principle is fulfilled.

Partial derivatives $\partial^{2}S/\partial Q_{\nu }^{2}$ describe the saturation of the system with the extensive variables, and $\partial^{2}S/\left(\partial Q_{\nu }\partial Q_{\kappa }\right)$ determines the substitution and complementation of the extensive variables in the macrosystem. If the extensive variables are substitutes, then an increase in one of them reduces the intensive variables adjoined with the other extensive variable. If the extensive variables are complements, then an increase in one of them, on the contrary, increases the intensive variables adjoined with the other extensive variable.

## 9. Metric Features of Entropy

Let us consider a particular case of exchange processes in a macrosystem consisting of two subsystems X and Y, where the fluxes linearly depend on the differences between the intensive variables of the subsystems:(24)$$q_{XY,\nu }\left(v_{X},v_{Y}\right)=\frac{{dQ}_{Y\nu }}{dt}=\sum_{\kappa =1}^{N}\alpha_{\nu \kappa }\varphi_{\kappa }\left(v_{X\kappa },v_{Y\kappa }\right),$$
where $\varphi_{\kappa }\left(v_{X\kappa },v_{Y\kappa }\right)=v_{Y\kappa }-v_{X\kappa }.$

**Proposition** **1. **
*Matrix $A=\left(\alpha_{\nu \kappa }\right)$
is the matrix of infrastructure coefficients describing the exchange possibilities at the boundary between subsystems and it is a symmetric matrix.*


**Proof.** Given the self-similarity property, the system can be divided into a statistically significant disjoint set of subsystems such that the characteristics of these subsystems form a representative sample of random variables $\tilde{q},\tilde{\varphi }$, etc. The eigenvectors of the matrices $Cov\left[\tilde{q}\right]$ and $Cov[\tilde{q},\tilde{\varphi }]$ coincide and form a system of orthonormal vectors. Since the eigenvectors of any matrix and its inverse are the same, $Cov\left[\tilde{q}\right]$ and $Cov[\tilde{q},\tilde{\varphi }]$ are symmetric and commutative, so their product is a symmetric matrix. Since $M[\tilde{q}]={Cov}^{T}[\tilde{q},\tilde{\varphi }]Cov\left[\tilde{q}\right]M\left[\tilde{q}\right]$, the matrix of phenomenological coefficients $A={Cov}^{T}[\tilde{q},\tilde{\varphi }]Cov\left[\tilde{q}\right]$ is symmetric (Onsager conditions). □

**Proposition** **2. **
*Matrix $A=\left(\alpha_{\nu \kappa }\right)$ is a positive definite matrix.*


**Proof.** The right part of Equation (16) under condition (24) has a quadratic form:
(25)$$\sigma (v_{X},v_{Y})=\varphi^{T}(v_{X},v_{Y})A\varphi \left(v_{X},v_{Y}\right).$$
For any positive values of the exchange driving forces $\varphi \left(v_{X},v_{Y}\right)$, the entropy production is positive, as shown in (15). Therefore, A is a positive definite matrix. It means that of the dependencies σ(φ) and σ(q) under the linear dependence of the flows on the driving forces are convex. □


The consequence of this proposition is that entropy production is a metric in the space of stationary processes and can be used to determine the distance between processes(26)$$\delta \left(a,b\right)=\sqrt{∆\sigma (v_{X},v_{Y})}=\sqrt{{\left(\varphi_{a}-\varphi_{b}\right)}^{T}A\left(\varphi_{a}-\varphi_{b}\right)}.$$

Let us present some properties of this metric:
–Zero in the space of stationary processes represents reversible processes for which σ=0. According to the third Levitin–Popkov axiom, there is the only reversible process in the macrosystem;–The distance between two processes a and b is determined as $\delta^{2}\left(a,b\right)={\left(\varphi_{a}-\varphi_{b}\right)}^{T}A\left(\varphi_{a}-\varphi_{b}\right)$. It is evident that $\delta \left(a,a\right)=0;\delta \left(a,b\right)=\delta \left(b,a\right)$;–The distance $\delta \left(a,b\right)$ satisfies the triangle inequality due to A being a positive definite symmetric matrix; all its eigenvalues λ are positive real numbers.


Thus, the entropy production in a macrosystem characterizes the distance of stationary exchange processes occurring in such a system from the corresponding reversible process. For linear systems, it is possible to consider generalized, averaged over probability to measure stationary processes: $\forall s\in R\exists \mu \left(s\right)=P(\sigma \left(v_{X},v_{Y}\right)=s):\;\bar{\sigma }=\int_{-\infty }^{+\infty }s\;\mu \left(s\right)ds$. Such an averaging for cyclic and stochastic processes is illustrated on Figure 3.

The distance between irreversible processes is an important concept for analyzing complex systems. The class of minimal dissipation processes indicates the limit of performance of macrosystems when the average intensities of the processes in it are restricted. To determine the effectiveness of an arbitrary process, it is necessary to find a distance between this process and the minimum dissipation process. This distance is determined by the production of entropy.

## 10. Trajectories of the Exchange Process

For non-stationary exchange processes, it is necessary to determine the trajectory of the process, i.e., the change in time of all subsystem parameters when approaching the equilibrium state. For a linear dependence between fluxes and driving forces (24), we derive a differential equation that determines the change in the driving forces of the exchange process in a macrosystem consisting of two subsystems X and Y. Under the given initial conditions $\varphi_{XY}\left(0\right)=\varphi_{0}$, this equation describes all the parameters of the subsystems.

Full differentials of functions $v_{i\kappa }\left(Q_{i}\right),i\in \left\{X,Y\right\},\kappa =1,\ldots ,N$ shall be written as(27)$$\forall \kappa =1,\ldots ,N:\sum_{\nu =1}^{N}Q_{\nu }\frac{\partial^{2}S}{\partial Q_{\nu }\partial Q_{\kappa }}=0.$$
Subtracting the equations for subsystem X from the equations for subsystem Y and taking into account the linear dependencies of fluxes on driving forces (24) and the fact that the driving forces are the differences between the corresponding intensive variables of the subsystems ($d\varphi_{XY}/dt=dv_{Y}/dt-dv_{X}/dt$), we obtain(28)$$\forall \kappa =1,\ldots ,N:\sum_{\nu =1}^{N}Q_{\nu }\frac{\partial^{2}S}{\partial Q_{\nu }\partial Q_{\kappa }}=0.$$

Equation (28), together with the initial conditions, determines the trajectory of the exchange process.

## 11. Conclusions

The metric properties of entropy production allow for the determination of both the class of minimally irreversible processes [11] and the quantitative distance between the processes. The stationarity restriction can be removed either by averaging the process parameters over time or due to the superposition principle for processes in linear systems. A generalized, independent-of-the-nature-of-the-processes, macrosystem model is useful for the formalization and investigation of extreme performances of complex, hierarchically related systems. In these macrosystem, there is a vector of entropy functions. It gives the macrosystem as additional degree of freedom at the level of subsystems.

The given proofs of the phenomenological properties of macrosystems are valid for thermodynamic systems but can also be applied to the systems of a different nature, particularly economic systems [12,13], information exchange systems in communication networks [14], and high-performance computers [15], where the number of computational cores is already comparable to the number of molecules in 1 μm^3^ of gas. The main difference in the description of macrosystems of different natures lies in the definition of the extensive quantities that describe the state of a system and in the relationships between the fluxes that arise from interactions between subsystems. In thermodynamic macrosystems, the extensive variables are internal energy, mol number, and volume [6]; in economic macrosystems, the extensive variables are resources, goods, and welfare [12]; in information macrosystems like recommendation systems, the extensive variables are database size and the number of exposed marks [16]; and so on. In information macrosystems, we restrict ourselves to describing syntactic information exchange [17]. Social macrosystems with semantic and pragmatic information exchange processes require a special conceptual apparatus, which is markedly different from the one usually used in formal logic [18]. For signal transmission systems, the extensive variables are the number of processors of a given type, the amount of memory, and the entropy properties of the computing power; the intensive variables are determined by the contribution to the increase in computing power that each type of hardware provides. When modelling algorithms as complex systems, the extensive variables can be used as control values, the entropy corresponds to the objective function, and the intensive variables are the values of the Lagrange multipliers. Economic analogies of complex systems are often considered. At the micro level, extensive quantities include the stock of goods while intensive quantities include the prices and the values of the goods. The welfare function has entropy properties in microeconomic systems. At the macro level, it is advisable to choose the gross regional product as the entropy, which is a function of vector of production factors. The intensive variables in this model are prices in real terms.

Entropy as an extensive quantity is introduced into different models of self-similar systems. Integration problems of the welfare function have been observed in [12]. The first-order homogeneity properties for gross regional product have been proven: the Cobb–Douglas production function is most commonly used in approximation. For communication networks, the entropy properties of the indicator of congestion have been proven [14]. These examples show the universality of a macrosystemic approach to modelling complex systems of different natures.

The graph of interaction between subsystems can be different for each type of nature of extensive variables. Such a complex system can be represented as a multigraph. Multigraphs where two graph nodes can be connected both by edges of different colours and by groups of edges of different colours, which correspond to multiple contact points between subsystems. The absence of edges between the multigraph’s nodes means that the nodes are isolated from each other. Edges of different colours corresponding to different extensive variables can be either co-directed or opposite.

The specific behaviour of individual elementary entities is especially relevant for social systems, where free and often unmotivated decision-making is possible and can complicate the application of the model. Nevertheless, the model can still be used to describe processes of various natures occurring simultaneously within a system. For example, a high-performance computer can be represented as a macrosystem in which hardware, software, and engineering subsystems exchange energy, signals, and information, and the average intensities of the computational processes and heat transfer processes are considered to be given. A computer does not perform mechanical work; therefore, all consumed electrical energy is converted into heat and must be dissipated into the surrounding environment.

## Figures and Tables

**Figure 1 entropy-27-01136-f001:**
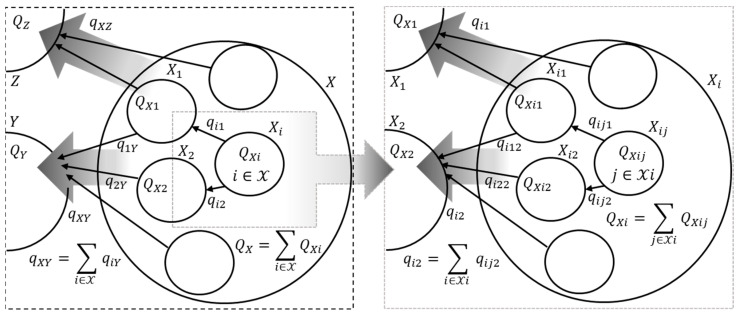
An example of the self-similar oriented weighted graph corresponding to a macrosystem. The nodes (the circles) correspond to the subsystems, and the edges (the arrows) correspond to the fluxes between the subsystems.

**Figure 2 entropy-27-01136-f002:**
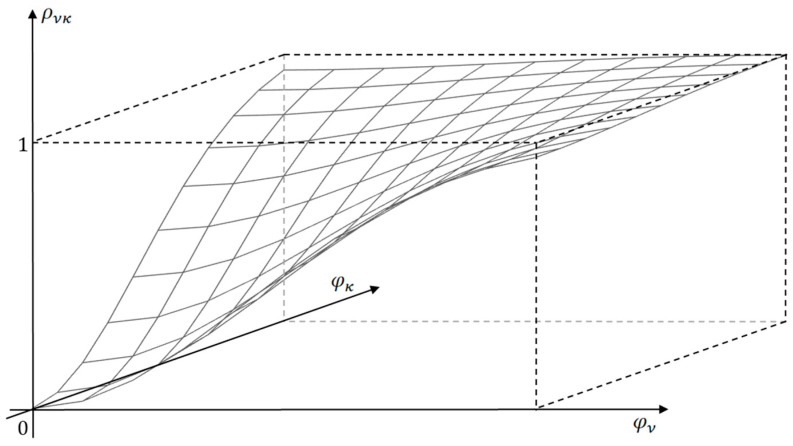
Dependency of correlation between fluxes $\rho_{\nu \kappa }\left(\varphi_{XY}\right)$ in a macrosystem: $\varphi_{\nu }$ and $\varphi_{\kappa }$ are the components of the vector $\varphi_{XY}$.

**Figure 3 entropy-27-01136-f003:**
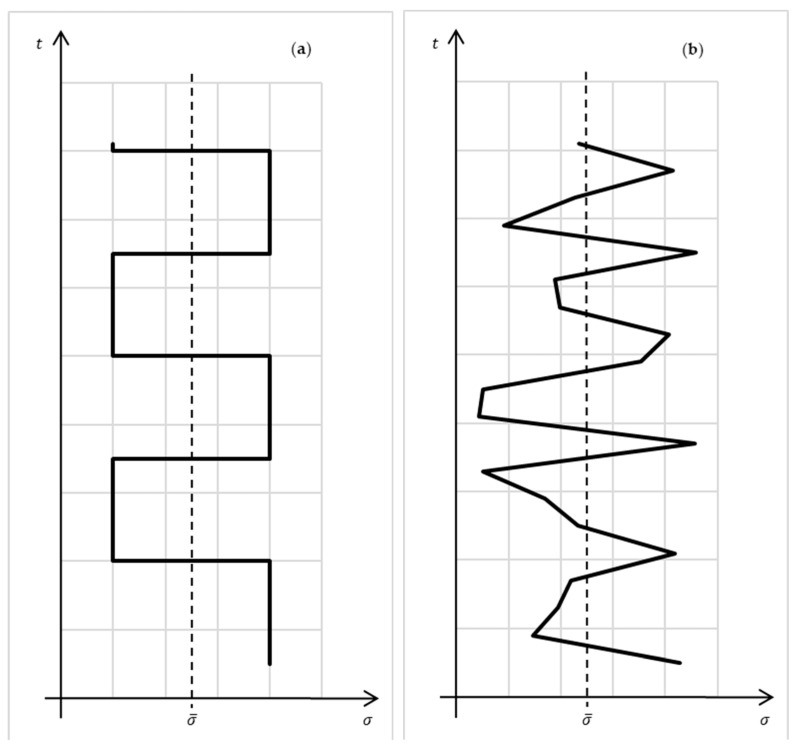
Averaging of entropy production in cyclic (**a**), and stochastic (**b**) stationary processes.

## Data Availability

No new data were created or analyzed in this study. Data sharing is not applicable to this article.

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
