# Peer review of "Thermodynamic Theory of Macrosystems: Entropy Production as a Metric"

_entropy, 2025, doi:10.3390/e27111136_

Round 1

Reviewer 1 Report

Comments and Suggestions for Authors
  1. The author often uses the term "stock of extensive variables". This is nothing more than the "value of an extensive variable", so the introduction of this term seems superfluous.
  2. In condition (1), it is not clear what the double index i means?
  3. Line 171. Scale neutrality does not need to be proven. The flows depend on intense variables, and they are homogeneous of the zero order and thus do not depend on scale.
  4. Line 192. "given fixed control ". If the control is set, it means it is fixed.
  5. Lines 280-282: "Among the stationary processes, we single out the classes of reversible processes and processes of minimal dissipation ." The intensity of reversible processes is constant and equal to zero, and the processes of minimal dissipation are usually not stationary.
  6. Lines 384-385: "The distance between processes is a vector"?? It is difficult to imagine that a metric in some space is a vector. How then is the triangle inequality formulated?? It is necessary to give at least one example.
  7. Conclusions are not sufficiently substantiated: for macrosystems of each nature, it is necessary to prove the existence of an analog of entropy in order to be able to extend properties similar to thermodynamic systems, economic, social, informational systems, and generally to macrosystems.

Author Response

I thank the reviewer for the questions. I tried to do my best to correct the text to make it more clear and improve it in accordance with the comments.

  1. The author often uses term "stock of extensive variables". This is nothing more than "value of an extensive variable", so introduction of this term seems superfluous.

I have corrected the terms in accordance with the comment.

  1. In condition (1) not clear what the double index i means?

There should not be such an index. To make it clear I added the clarification “pairwise disjoint” wherever doubts may arise.

  1. Line 171. Scale neutrality does not need to be proven. The flows depend on intense variables, and they are homogeneous of the zero order and thus do not depend on scale.

I eliminated that questionable phrase.

  1. Line 192. "given fixed control ". If control is set, it means it is fixed.

I understand that this phrase can be excess, but I can not change it because it is citation of the Levitin – Popkov Axioms (Levitin, E. S.; Popkov, Yu. S. Axiomatic Approach to Mathematical Macrosystems Theory with Simultaneous Searching for Aprioristic Probabilities and Stochastic Flows Stationary Values. Proceedings of the Institute for System Analysis RAS 2014, vol. 64 (3), pp. 35 – 40; reference [9] in the paper).

  1. Lines 280-282: "Among the stationary processes, we single out the classes of reversible processes and processes of minimal dissipation." The intensity reversible processes constant and equal zero, and processes of minimal dissipation are usually not stationary.

The first part is correct and it is mentioned in the text. The second part is partially right. For example, in Carnot cycle we observe stationary process of minimal dissipation. It depends of the features of the heat source. In any case when the driving force is linear and therefore the superposition principle is being applied, we can introduce a class of the minimal dissipation processes not only for stationary processes. This text is corrected in accordance with the comment.

  1. Lines 384-385: "The distance between processes is a vector"?? It is difficult to imagine that a metric in some space is a vector. How then is the triangle inequality formulated?? It is necessary to give at one example.

When we consider a system with interacting objects of different natures we need introduce a vector of entropy functions. It corresponds to Pareto optimum of stationary state and can be considered as a vector of distances. All elements of the vector must satisfy properties of distance: symmetry and the triangle inequality. I think that as a first approximation it can be envisioned as vector of distances (L2, Lp) in ordinary space. The text is corrected in accordance with the comment.

  1. Conclusions are not sufficiently substantiated: for macrosystems of each nature, it is necessary to prove the existence of an analog of entropy in order to be able to extend properties similar to thermodynamic systems, economic, social, informational systems, and generally to macrosystems.

I added some links to the proofs that the features of entropy are correct for interacting systems of different natures. That is why a generalized theory of macrosystem is relevant for research of complex systems.

Reviewer 2 Report

Comments and Suggestions for Authors

This paper presents a very interesting generalization of the concepts in thermodynamics to other systems. It provides the mathematical structure to use the analogs of linear Onsager theory for those systems described in the conclusion section. The presentation is overall clearly written and can be published after a number of minor points have been taken care of:

Line 11:

The concept of self-similarity vs hierarchical structure should be explained in the context of figure 1; in addition the notion “coloured” seems strange as figure 1 is not coloured.

Line 31

“In thermodynamic systems the elementary particles are molecules”: this is not true in general, thermodynamics does not need discrete particles as a concept or a minimal system size.

Line 69

“… satisfy the conservation law”: what about chemical substances which are CREATED or DESTROYED in chemical reactions?

Line 72

“mass” is usually NOT a thermodynamic variable – usually it is mol number (amount of substance – if there are more than one then several amounts)

In the text the term STOCK is used with the same meaning: please unify the nomenclature.

Line 115

“different colours”: please define the meaning of COLOUR.

Line 121

Add colour to the figure

Line 136

Provide a thermodynamics based argument why the distribution of flux sizes should be a multivariate normal distribution – alternatively just state that as condition which must be satisfied.

Line 171

Not the variables have to satisfy the neutral scale effect conditions (homogeneity) – it is the system with its entropy function which has to satisfy that.

Line 182

“objective function” is used as a term: please define what it means

Line 189

What is the “admisissible set” here and what is the connection to the text before?

Line 242 ff

Is entropy now the Shannon entropy of the distribution function?

The distribution function which is generated by maximizing the Shannon entropy under the condition of certain mean values is NOT a Gaussian – it is an exponential. So the Gaussian is NOT maximizing entropy.

Line 260

The figure needs explanation of what it should convey to the reader!!!!

Line 286

This is a cryptic statement: what does it mean?

Line 344

“ …. The entropy production is positive”  Is this a statement of fact and is thus used as an additional input needed OR is this a consequence of the previous presentation?

Author Response

I thank the reviewer for the questions. I tried to do my best to correct the text to make it more clear and improve it in accordance with the comments.

  1. Line 11: The concept of self-similarity vs hierarchical structure should be explained in the context of figure 1; in addition the notion “coloured” seems strange as figure 1 is not coloured.

I have corrected the terms in accordance with the comment.

  1. Line 31 “In thermodynamic systems the elementary particles are molecules”: this is not true in general, thermodynamics does not need discrete particles as a concept or a minimal system size.

It is true, but to explain macroscopic physical properties we use statistical mechanics, especially to study the simplest non-equilibrium situation of a steady state current flux in a system of many particles.

I have corrected the text in accordance with the comment.

  1. Line 69 “… satisfy the conservation law”: what about chemical substances which are CREATED or DESTROYED in chemical reactions?

There are some conservation laws in chemistry too. I am not ready to use the proposed approach to chemical reactions but I am sure that it is possible to form the interaction graph for this case too. In any case self-similarity property is valid to chemical reactions. Some conservation laws are true in this case: they are law of conservation of mass, the law of definite proportions (i.e., the law of constant composition), the law of multiple proportions and the law of reciprocal proportions. Chemical reactions can neither create nor destroy matter.

  1. Line 72 “mass” is usually NOT a thermodynamic variable – usually it is mol number (amount of substance – if there are more than one then several amounts)

I have corrected the terms in accordance with the comment.

  1. In the text the term STOCK is used with the same meaning: please unify the nomenclature.

I changed "stock of extensive variables" to "value of an extensive variable" to correct the terms.

  1. Line 115 “different colours”: please define the meaning of COLOUR.

Colour means extensive variable exchanged between the subsystems. It can be shown as vector of interaction, so I eliminated the concept of colour in the main part of the paper. However when we need to describe the interaction in a system, where extensive variables have different natures, it makes sense to mention colour concept because subgraphs of different colours can be different.

That is why concept of multigraph is displaced to the conclusions.

  1. Line 121 Add colour to the figure

It is not necessary because multigraphs are no longer discussed in this part of the paper.

  1. Line 136 Provide a thermodynamics based argument why the distribution of flux sizes should be a multivariate normal distribution – alternatively just state that as condition which must be satisfied.

Distribution of flux sized should be Gaussian due to maximum entropy in equilibrium state. If variance is given (or covariance matrix is given) the maximum corresponds to Gaussian distribution.

  1. Line 171 Not the variables have to satisfy the neutral scale effect conditions (homogeneity) – it is the system with its entropy function which has to satisfy that.

The system at issue is a macrosystem, where any subsystem is characterized by extensive and intensive variables. Due to entropy function allows one to construct the model of the macrosystem. The neutral scale effect condition provides self-similarity effect.

  1. Line 182 “objective function” is used as a term: please define what it means

Objective function means “optimality criterion” which should be maximized. It is a common term, so I did not changed it.

  1. Line 189 What is the “admisissible set” here and what is the connection to the text before?

It is the misprint, of course. I fixed this bug. In that part of the paper the Levitin–Popkov Axioms were cited. The permissible set is the part of the citation.

  1. Line 242 ff Is entropy now the Shannon entropy of the distribution function?

In article it is shown that entropy meets the conditions of Shannon entropy if the number of subsystems is large enough and covariation matrix is given. I do not change the concept of entropy.

  1. The distribution function which is generated by maximizing the Shannon entropy under the condition of certain mean values is NOT a Gaussian – it is an exponential. So the Gaussian is NOT maximizing entropy.

Fortunately, entropy of Gaussian distribution is more than entropy of exponential distribution. For Gaussian distribution  when for exponential distribution entropy is  only.

  1. Line 260 The figure needs explanation of what it should convey to the reader!!!!

I added some additional explanations to the text and to the figure 2 caption in accordance with the comment.

  1. Line 286 This is a cryptic statement: what does it mean?

It was too strong: I changed it in accordance with the comment.

  1. Line 344 “ …. The entropy production is positive” Is this a statement of fact and is thus used as an additional input needed OR is this a consequence of the previous presentation?

It is a consequence of the previous presentation, it follows from (15). I have corrected the text in accordance with the comment.

Round 2

Reviewer 1 Report

Comments and Suggestions for Authors

It's good that the author agrees with the comments.

It remains to do the following: 1. The author will add an example of a specific problem in which his terminology allows us to obtain a previously unknown solution. 2. The author will show in detail how his approach can be extended to a system whose nature differs from the thermodynamic one. And what new opportunities follow from this. 3. The author will clarify how entropy production determines not only the distance of a process from a reversible one, but also the distance between two irreversible processes. 4. The author will remove the concept of vector distance, as it does not stand up to any criticism.

Author Response

1. The author will add an example of a specific problem in which his terminology allows us to obtain a previously unknown solution. 

An example of a specific problem in which the proposed approach allows to obtain new solutions is the task of minimizing energy consumption by high-performance computers. The problem is related to the determination of the optimal mode of operation both in calculations used millions of cores, and in engineering system that provides efficient cooling of the computer. The use of the proposed approach has allowed to develop a high-performance computer, where the unproductive energy consumption does not exceed 1% of the heat emission of the processors. However, this particular problem formulization and the result obtained are outside of the topic of the paper. The aim of this study is to construct an axiomatic theory of macrosystems, in which phenomenological properties are proved from concrete assumptions about the structure of a macrosystem.

2. The author will show in detail how his approach can be extended to a system whose nature differs from the thermodynamic one. And what new opportunities follow from this. 

The proposed approach can be used not only to describe systems other than thermodynamic, but also multimodal systems where exchange processes have different nature. For above example, energy and information exchange processes are observed in the high-performance computers. Optimal mode of operation of the software and optimum mode of operation of the cooling system (both in the sense of minimizing entropy production) allow to solve the problem of building a high-performance computing complex, with maximum computational reliability and minimum total power consumption.

However, the detailed formulation and solution of this task is the subject of a separate study that is close to being finalized and will necessarily be published.

3. The author will clarify how entropy production determines not only the distance of a process from a reversible one, but also the distance between two irreversible processes. 

The distance between irreversible processes is an important concept for analyzing complex systems. The class of minimal dissipation processes indicates the limit of performance of macrosystems when the average intensities of the processes in it are restricted. To determine the effectiveness of an arbitrary process, it is necessary to find a distance between this process and the minimum dissipation process. This distance is determined by the production of entropy. The description of this calculation is added to the article.

4. The author will remove the concept of vector distance, as it does not stand up to any criticism.

The author has removed the reference to the vector of distances, as this is not a key issue for this study.

Round 3

Reviewer 1 Report

Comments and Suggestions for Authors

No comments